# SAW Chemical Array Device Coated with Polymeric Sensing Materials for the Detection of Nerve Agents

**DOI:** 10.3390/s20247028

**Published:** 2020-12-08

**Authors:** Jinuk Kim, Hyewon Park, Jihyun Kim, Byung-Il Seo, Joo-Hyung Kim

**Affiliations:** 1Laboratory of Intelligent Devices and Thermal Control, Department of Mechanical Engineering, INHA University, inha-ro 100, Incheon 22212, Korea; 22191015@inha.edu (J.K.); hyewon.park@inha.edu (H.P.); 2Inha Institute of Space Science and Technology (Inha IST), INHA University, inha-ro 100, Incheon 22212, Korea; jihyunn.kim@inha.ac.kr (J.K.); siry0305@naver.com (B.-I.S.)

**Keywords:** SAW array sensor, chemical warfare agent, surface acoustic wave, polyhedral oligomeric silsesquioxane, dimethyl methylphosphonate

## Abstract

G nerve agents are colorless, odorless, and lethal chemical warfare agents (CWAs). The threat of CWAs, which cause critical damage to humans, continues to exist, e.g., in warfare or terrorist attacks. Therefore, it is important to be able to detect these agents rapidly and with a high degree of sensitivity. In this study, a surface acoustic wave (SAW) array device with three SAW sensors coated with different sensing materials and one uncoated sensor was tested to determine the most suitable material for the detection of nerve agents and related simulants. The three materials used were polyhedral oligomeric silsesquioxane (POSS), 1-benzyl-3-phenylthiourea (TU-1), and 1-ethyl-3-(4-fluorobenzyl) thiourea (TU-2). The SAW sensor coated with the POSS-based polymer showed the highest sensitivity and the fastest response time at concentrations below the median lethal concentration (LCt_50_) for tabun (GA) and sarin (GB). Also, it maintained good performance over the 180 days of exposure tests for dimethyl methylphosphonate (DMMP). A comparison of the sensitivities of analyte vapors also confirmed that the sensitivity for DMMP was similar to that for GB. Considering that DMMP is a simulant which physically and chemically resembles GB, the sensitivity to a real agent of the sensor coated with POSS could be predicted. Therefore, POSS, which has strong hydrogen bond acid properties and which showed similar reaction characteristics between the simulant and the nerve agent, can be considered a suitable material for nerve agent detection.

## 1. Introduction

Chemical Warfare Agents (CWAs) are classified into nerve, blood, choking, and blister agents according to the effects they have on the body and their intended use. Nerve agents such as sarin (GA), tabun (GB), soman (GD), and VX are among the most toxic, and usually contain organophosphates [1,2,3]. The toxicity of these nerve agents can be quantitatively expressed using the median lethal concentration (LCt_50_) expressed in $\mathrm{mg}\cdot \min/m^{3}$. The LCt_50_ of GA is 400 $\mathrm{mg}\cdot \min/m^{3}$, and that of GB is 100 $\mathrm{mg}\cdot \min/m^{3}$; exposure at these concentrations can cause serious damage to humans [1]. The production and use of CWAs are limited by the Organization for the Prohibition of Chemical Weapons (OPCW), but these compounds could potentially still be used in warfare and terrorism, posing a threat not only to military personnel, but also to civilians [1]. Therefore, early detection is key. Generally, nerve agents can be detected using various techniques such as photoionization [4], ion mobility spectroscopy [5,6], an electric nose [7,8], flame photometry [9,10,11], and IR spectroscopy [12,13], but most detectors using these methods are heavy, bulky, and consume lots of power.

Surface acoustic wave (SAW) sensors, a kind of the chemical detector, are sensitive, small, and require extremely low power to operate [14,15,16,17,18]. As such, they have great potential for use as handheld detectors. The sensing material on the surface of the delay line of the SAW sensor is one of the most important components to selectively detect CWA vapors, because it has a large impact on the mass loading effect or conductivity changes, which are the prime sensing principles of SAW sensors [19,20]. Hydrogen bond acidic polymers such as fluoropolyol, fluoroalcohol polysiloxane, and fluoroalcoholic linear polysiloxane have been utilized as sensing materials for the detection of nerve agents in security and defense applications because they can selectively interact with the target analytes by hydrogen bonding [21]. These hydrogen-bond acidic polymers contain hydroxyl derivatives including alcohol, carboxylic acid, or phenolic groups, and the addition of electron withdrawing halogens in close proximity to hydroxyl groups yields a dramatic increase in hydrogen bond acidity with a concomitant reduction in hydrogen bond basicity [21,22,23,24].

In this paper, SAW sensors coated with polyhedral oligomeric silsesquioxane (POSS), 1-benzyl-3-phenylthiourea (TU-1), and 1-ethyl-3-(4-fluorobenzyl)thiourea (TU-2), synthesized by Jang [25] and Song [26] in a laboratory in Korea, were tested to investigate their sensitivity to GA, GB, and dimethyl methylphosphonate (DMMP). These sensors were tested simultaneously using a portable SAW array device which included one uncoated SAW sensor to reduce the effect of moisture and three coated SAW sensors under the same environmental conditions.

## 2. Experiment

### 2.1. Design of SAW Array Sensor

The SAW sensor module is shown in Figure 1a. The SAW sensors are based on piezoelectric materials, influencing their characteristics. The most commonly used piezoelectric materials for SAW sensors are quartz, lithium niobate (LiNbO_3_), and lithium tantalate (LiTaO_3_) [27]. Different wave types are obtained depending on the piezoelectric material, crystal cut, and temperature dependence [28]. While LiNbO_3_ and LiTaO_3_ show higher frequency change by temperature variation, ST-cut quartz has the advantage of thermal stability. The sensitivity of the sensor based on the choice of piezoelectric material is affected by $f_{c}=v/\lambda$, which is the center frequency of the SAW sensor, where λ, v, and $f_{C}$ are wavelength, SAW velocity, and center frequency, respectively [29,30,31]. Figure 1b,c show optical microscope images of the SAW delay line. The SAW sensors were fabricated on a ST-cut quartz substrate having an aluminum interdigit transducer (Al-IDT) structure designed for a 250-MHz center frequency. Al electrodes were deposited on the substrate and titanium was used as an adhesion metal for the deposition of the IDTs. The center frequency was calculated as 250 MHz because the SAW velocity on the ST-cut quartz substrate was 3158 m/s and the wavelength of the Al electrode was 12.632 μm.

The SAW array device was constructed in a four-channel configuration consisting of reference and sensing channels, as shown in Figure 2a. The uncoated SAW sensor improved the performance of the SAW array device by compensating for the frequency shift caused by environmental effects. Each of the three sensors was coated with different sensing materials on the delay lines. Each channel to output the frequency signal was made up of a low pass filter (LPF), an amplifier, and a SAW sensor. The coated SAW sensors from channels 1–3 are for detecting vapors, while the uncoated sensor on channel 4 is a reference sensor. The frequency signals of the coated SAW sensors are cross-coupled with the signal of the reference sensor to transmit the difference between the two signals [32]. Then, real-time measurement signals generated by each channel are transmitted to the display via the multiplexer (MUX).

Figure 2b shows the relevant circuit-board design with the portable battery, which is divided into two parts: the oscillator board and the digital-signal processing board. The fourth sensor is the reference sensor and the others are coated sensors. Each sensor is mounted on each channel on the oscillator board. The board with mounted SAW sensors is connected to the digital-signal processing board. This SAW array device is powered by connecting the portable battery.

### 2.2. Materials Synthesis and Coating Process

POSS and thiourea (TU) -based polymers are used as sensing materials to detect GA, GB, and D MMP (Sigma Aldrich, MO, USA), as shown in Figure 3. GA and GB were supplied by the Netherlands Organization for applied scientific research (TNO, Rijswijk, Netherland). These were obtained from previously synthesized stocks of TNO. The POSS-based polymer was synthesized using 4-(Trifluoromethyl) phenol and PSS-Octakis (dimethylsilyloxy) substituted through Karstedt’s catalyst [25]. TU-1 was synthesized using a TU-based polymer and N-benzyl groups attached to both sides of the TU base. The N-benzyl groups successfully reduced the self-aggregation of TU molecules, and the additional CH − π interaction strengthened the binding of TU and organophosphonate. TU-2 was synthesized using a TU-based polymer and 4-fluoro-benzyl, as this enhances the hydro bonding by altering the electron density of the N-benzyl group via electron-withdrawing [26].

The POSS and TU-based polymers were diluted to a 1:1 concentration using ethanol and dimethylformamide as solvents, respectively. These three solutions were dispersed with a sonicator for 1 h, and then 0.5 μL of these solutions was coated on the surface of the delay line of the SAW sensors by a drop coating process. Finally, these sensors were heated to 60 °C for 2 h in a convection oven [32].

### 2.3. Experimental Apparatus

This experiment was conducted in two ways, depending on the detection of nerve agents or the simulant, because nerve agents cannot be handled in a typical laboratory. The experiment detecting nerve agents took place at the High Tox laboratory of TNO, while experiments detecting the simulant were conducted in our laboratory in South Korea. The SAW array device was evaluated in a test facility at TNO that has a dynamic vapor generation system with a canister, as shown in Figure 4. The needle of the syringe filled with the agents was inserted into the heated injector (140 °C for GB and 175 °C for GA) of a heat chamber. The agents were evaporated in the injector and mixed with nitrogen ($N_{2}$) gas delivered at a flow rate or 1 L/min in the heat chamber. A set of valves controlled the exposure condition of the SAW array device. Figure 4a shows the main mechanism of the experiment when the device was exposed to CWAs. The vapor concentration of the CWAs was controlled by changing the volume flow rate of the syringe driver. The flow rate of GA was changed from 10 nL/min to 100 nL/min, and that of GB from 5 nL/min to 500 nL/min. After the exposure time, the agent was released to the outside through the canister, and pure $N_{2}$ gas was injected into the SAW array device as shown Figure 4b.

A gas feeding system was used to test the SAW array device shown in Figure 5. It consisted of stainless-steel tubes through which the gases flowed, check valves that allowed vapor to flow in only one direction, two mass-flow controllers (KOFLOG, Japan) to control the concentration of the gases, and a readout unit. A bubbler was mounted on the carrier gas line, and the target material was maintained at 25 °C in a two-necked-flask using a constant-temperature water bath. $N_{2}$ gas was used as both the dilution and carrier gas. The $N_{2}$ gas entered the two-necked flask at a constant flow rate maintained by the mass-flow controllers, and DMMP, the target material, was released into the vapor by the vapor pressure [30,32].

## 3. Results

### 3.1. Sensitivity to GA Vapor

The response of the SAW array device to GA in the range of 50–100 $\mathrm{mg}/m^{3}$ (7.5–15 ppm) is shown in Figure 6. Upon exposure to GA vapor, the frequency response immediately changed, showing a saturated signal after a constant frequency shift according to the vapor concentration. The frequency shift of the SAW-1 showed a change of 11.0 kHz for GA at a concentration of 50 $\mathrm{mg}/m^{3}$. There was some data loss, shown in Figure 6 as a dash line, in the initial response at 50 $\mathrm{mg}/m^{3}$, but this did not affect the determination of an equilibrium state of adsorption. The 80% response time, *T*_80_, was defined as the time interval for the frequency shift reaching 80% of the maximum frequency to reduce the effect of noise from the equilibrium state of the sensors. *T*_80_ was about 50 s for GA at 100 $\mathrm{mg}/m^{3}$, and finally achieved a maximum frequency of 18.9 kHz. The frequency shifts of the SAW-2 and SAW-3 changed to 1.7 kHz and 2.8 kHz at a concentration of 50 $\mathrm{mg}/m^{3}$, respectively, which represented frequency shifts which were lower than the SAW-1. The SAW-2 or SAW-3 showed unstable signals in the equilibrium state upon exposure to a concentration of 100 $\mathrm{mg}/m^{3}$ due to their small frequency shifts relative to noise.

### 3.2. Sensitivity to GB Vapor

Figure 7 describes the response of each SAW sensor upon exposure to GB in the range of 5.5–550 $\mathrm{mg}/m^{3}$(0.95–96 ppm). The SAW-1 showed a frequency shift of 0.495 kHz at the lowest concentration of 5.5 $\mathrm{mg}/m^{3}$ which increased with increasing concentration. The frequency shift was represented as 13.7 kHz at 550 $\mathrm{mg}/m^{3}$, and the $T_{80}$, an average of $T_{80}$, was 45 s in the range of concentration. The SAW-2 indicated no significant frequency shift up to 55 $\mathrm{mg}/m^{3}$. The frequency shift of the sensor was 0.59 kHz at a concentration of 110 $\mathrm{mg}/m^{3}$, which was the limit of detection of SAW-2, i.e., 4.3 kHz at 550 $\mathrm{mg}/m^{3}$. The SAW-3 showed a frequency shift of 0.39 kHz from the lowest concentration but could not distinguish the signals at concentrations below 110 $\mathrm{mg}/m^{3}$. The observed frequency shifts were 0.64 kHz for 110 $\mathrm{mg}/m^{3}$ and 6.1 kHz for 550 $\mathrm{mg}/m^{3}$. It could be assumed that the reaction patterns of SAW-2 and SAW-3 would be similar upon exposure to GB because the sensors were made using the same TU-based polymer. The response patterns presented a spike because the sensors responded slowly to GB, i.e., they needed about 3 min to reach an equilibrium state. The baseline shifts of each sensor were caused by slower desorption than sorption in the concentration range over 110 $\mathrm{mg}/m^{3}$.

### 3.3. Sensitivity to DMMP Vapor

The dynamic responses of three SAW sensors coated with POSS, TU-1, and TU-2 to DMMP in a concentration range of 63 to 507 $\mathrm{mg}/m^{3}$ (12.5–100 ppm) are shown in Figure 8. One test cycle consisted of an exposure time of 3 min to DMMP vapor and a purging time of 5 min using 99.9% $N_{2}$ gas. The same test cycle was carried out once again to confirm the reproducibility of the sensors to exposure to DMMP vapor. The SAW-1 presented a frequency shift of 12.6 kHz at 507 $\mathrm{mg}/m^{3}$ and a stable frequency shift of 4.1 kHz at 63 $\mathrm{mg}/m^{3}$. The $T_{80}$ was 22 s at high concentration (507 and 381 $\mathrm{mg}/m^{3}$), but slowed to about 51 s at low concentration (254, 127, and 63 $\mathrm{mg}/m^{3}$). The frequency shifts of the SAW-2 and the SAW-3 were 14.6 kHz and 2.5 kHz at 507 $\mathrm{mg}/m^{3}$, respectively, and were unstable at 63 $\mathrm{mg}/m^{3}$. The frequency shift of the SAW-2 was greater than that of SAW-1 at the highest concentration, but lower than that of SAW-1 with decreasing concentrations of DMMP. The $T_{80}$ of SAW-2 and SAW-3 was approximately 83 s and 88 s, respectively.

### 3.4. Regression to Frequency Response of the Sensors

The difference of frequency, ∆f, in kHz before and after exposure to vapors can be expressed as a function of concentration, x, in $\mathrm{mg}/m^{3}$ of the vapor types, as described in Figure 9. Table 1 shows the slope, intercept, and R-square of these functions. All correlation coefficients were above 0.9. The slopes of GB and DMMP similarly appeared for SAW-1, which had high sensitivity with low concentrations of GA. The reactions to GB and DMMP presented a different trend, even though SAW-2 and SAW-3 were made by coating the sensing materials with the same TU-based polymer. The slopes of GB were smaller for SAW-2 and larger for SAW-3 than the slope of DMMP. Also, SAW-2 had the lowest sensitivity to the vapors except for DMMP within the overall concentration range.

The SAW sensor detected the G nerve agents and their simulants by changing the hydrogen bond acidity of the polymer to maximize the mass loading effect [33]. A particularly effective means of enhancing the hydrogen-bond acidity of the polymer is to incorporate fluorinated alcohol or fluorinated phenolic functional groups as substituents in the polymer structure [33,34,35]. It seems that the fluorinated phenolic functional groups of the POSS-based polymer reacted with the phosphonate oxygen group in GA, GB, or DMMP to form hydrogen bonds [25]. In contrast, TU-based polymers form hydrogen bonds between the NH groups of the polymers and the phosphonate oxygen groups to detect G nerve agents and their simulants [26]. Although all three sensing materials formed hydrogen bonds with the analytes, it was expected that the POSS-based polymer would show higher sensitivity than the other sensing materials because its fluorinated phenolic group maximizes the hydrogen bond acidity of the polymer by the electron-withdrawing effect of the fluorine atoms [33].

From the comparison of these sensitivities to the analyte vapors, we also confirmed that the frequency response to DMMP was similar to that for GB. DMMP is a simulant which physically and chemically resembles GB [36].

### 3.5. Long-Term SAW Response: Reliability of the POSS as a Sensing Material

Maintaining the performance of sensors for a long period is related to the reliability of the sensor. The stability of the SAW sensor coated with POSS was evaluated by assessing the increase of concentration at 0, 90, and 180 days after fabrication, as shown in Figure 10. One microliter of a mixture of POSS and ethanol at a ratio of 1:1 (mg:mL) was coated onto the 250 MHz SAW sensors. The response was measured using a vector network analyzer (MS46122A, Anritsu). The sensors were tested in a DMMP concentration range of 127–507 $\mathrm{mg}/m^{3}$ (25–100 ppm), and later retested in the same conditions. According to the results, it is hypothesized that the vapor concentration affects the performance of the sensors, making them more stable at higher concentrations. When comparing the performance of the sensor between the first and 180th days after fabrication, the level of reaction was maintained at 79% at the highest concentration, i.e., 507 $\mathrm{mg}/m^{3}$, and 64% at the lowest, i.e., 127 $\mathrm{mg}/m^{3}$. Performance degradation was mainly observed 0 to 90 days after fabrication, with only 10% or less occurring over days 90 to 180. This suggests that a sensor coated with POSS could maintain good long-term performance if initial performance degradation is prevented.

## 4. Conclusions

We designed and fabricated four 250 MHz SAW sensors coated with POSS (SAW-1), TU-1(SAW-2), and TU-2 (SAW-3), and evaluated their working performance by measuring the frequency shift upon exposure to GA, GB, and DMMP. In a response test to GA exposure within a concentration range of 50–100 mg/m^3^, all SAW sensors could detect the vapor at 50 $\mathrm{mg}/m^{3}$. For GB in the range of 5.5–550 $\mathrm{mg}/m^{3}$, SAW-1 and SAW-3 were effective at 5.5 $\mathrm{mg}/m^{3}$, the lowest concentration of GB, while the limit of detection of SAW-2 was shown to be 110 $\mathrm{mg}/m^{3}$. The response pattern of SAW-1 is a rectangular shape, but those of SAW-2 and SAW-3 presented spikes, indicating that the response to GB did not reach equilibrium during the exposure time. Also, SAW-2 and SAW-3 showed similar response patterns for the GB exposure because their sensing materials were synthesized with the same TU-based polymer. A response experiment for DMMP, a simulant of G nerve agents, was implemented in the range of 63–507 $\mathrm{mg}/m^{3}$. The SAW-1 showed high sensitivity to low concentrations of DMMP, while SAW-2 showed high sensitivity under the highest concentration, which gradually decreased as the concentration declined, ending with an unstable signal at the lowest concentration. The sensitivity of SAW-3 was the lowest among the three sensors for DMMP vapor. The results of the exposure test confirmed that among the three sensing materials, POSS showed the highest sensitivity, the fastest response time to G nerve agents, and the largest frequency response at concentrations of GA and GB below LCt_50_. The sensitivity of SAW-1 to a real agent could be predicted using the simulant, i.e., DMMP, a compound which physically and chemically resembles GB.

## Figures and Tables

**Figure 1 sensors-20-07028-f001:**
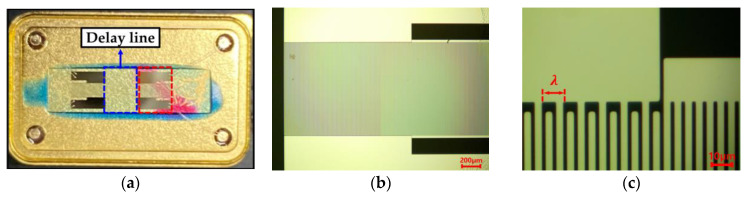
Optical microscope image of the IDT on a SAW sensor module: (**a**) SAW sensor module; (**b**) IDT pattern; (**c**) Design parameter of IDT.

**Figure 2 sensors-20-07028-f002:**
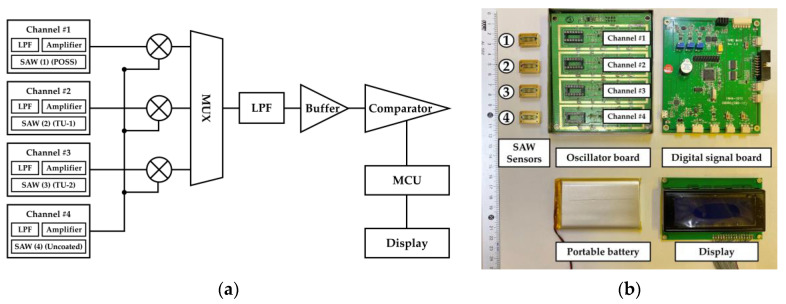
Four-channel SAW monitoring system: (**a**) Signal processing diagram; (**b**) Structure of the SAW array device with a portable battery. The insertion numbers indicate four different SAW sensors.

**Figure 3 sensors-20-07028-f003:**
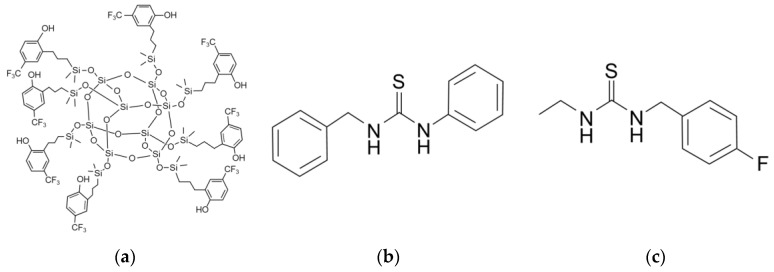
Chemical structure of the sensing materials: (**a**) POSS; (**b**) TU-1; and (**c**) TU-2 [32].

**Figure 4 sensors-20-07028-f004:**
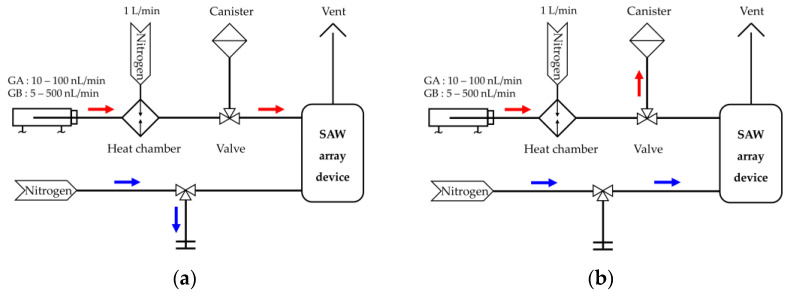
Schematic diagram of the experimental setup for testing the SAW array device with nerve agents (TNO, Netherland): (**a**) exposure to G nerve agents; (**b**) exposure to $N_{2}$ gas for purging.

**Figure 5 sensors-20-07028-f005:**
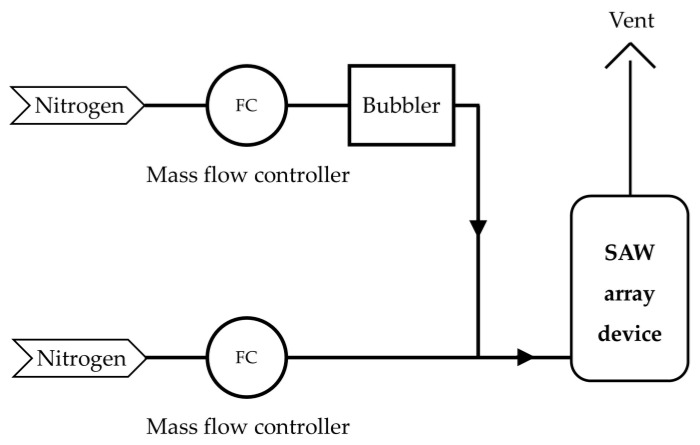
Schematic diagram of the in house experimental setup for testing the SAW array device with a simulant, DMMP (Inha University, South Korea).

**Figure 6 sensors-20-07028-f006:**
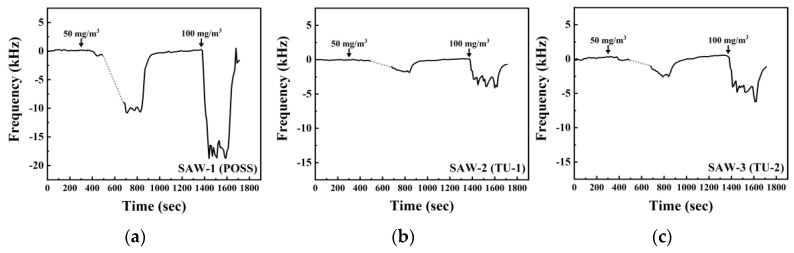
Dynamic response of the SAW sensors coated with different sensing materials: (**a**) POSS; (**b**) TU-1; and (**c**) TU-2 upon exposure to GA concentrations ranging from 50 to 100 $\mathrm{mg}/m^{3}$. The inserted numbers are the different concentrations of GA.

**Figure 7 sensors-20-07028-f007:**
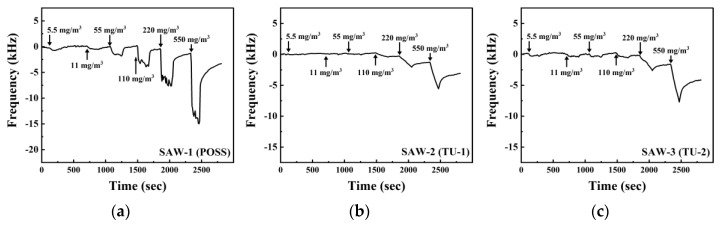
Dynamic response of the SAW sensors coated with different sensing materials: (**a**) POSS; (**b**) TU-1; and (**c**) TU-2 upon exposure to GB at concentrations from 5.5 to 550 $\mathrm{mg}/m^{3}$. The inserted numbers are the different concentrations of GB.

**Figure 8 sensors-20-07028-f008:**
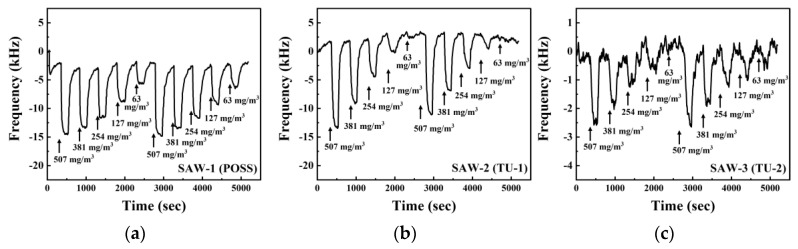
Dynamic response of the SAW sensors coated with different sensing materials: (**a**) POSS; (**b**) TU-1; and (**c**) TU-2 under the exposure to DMMP concentrations ranging from 63 to 507 $\mathrm{mg}/m^{3}$. The inserted numbers are the different concentrations of DMMP.

**Figure 9 sensors-20-07028-f009:**
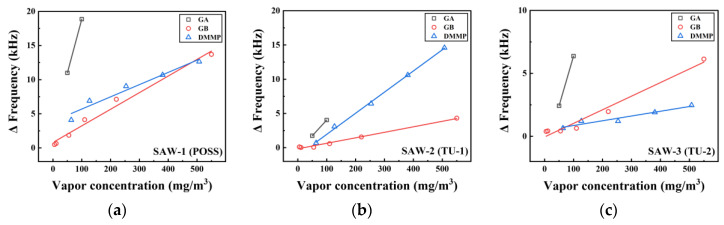
Response of the SAW sensors with different sensing materials of (**a**) POSS; (**b**) TU-1; and (**c**) TU-2 depending on the vapor type.

**Figure 10 sensors-20-07028-f010:**
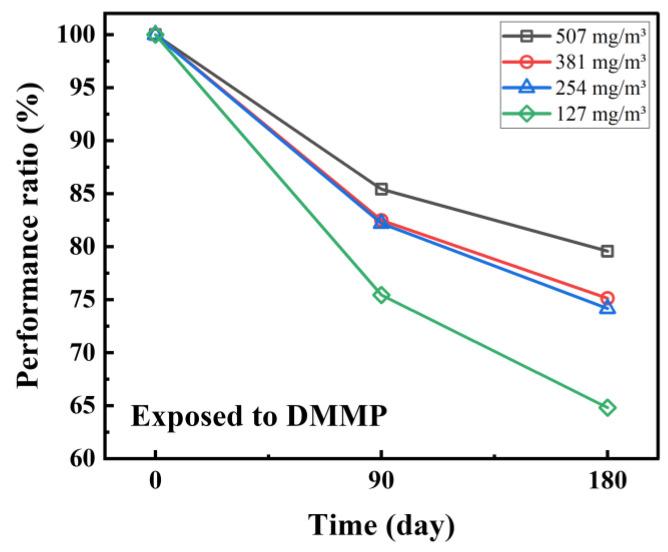
Performance over time of the SAW sensor coated with POSS.

**Table 1 sensors-20-07028-t001:** Linear parameters of the function ∆f=ax+b: *a*, *b*, and *R*^2^ are slope, intercept, and coefficient of determination, respectively.

	*a*	*b*	*R* ^2^
	**SAW-1 (POSS)**		
GA	0.157	3.118	-
GB	0.024	0.803	0.984
DMMP	0.018	3.906	0.957
	**SAW-2 (TU-1)**		
GA	0.046	−0.563	-
GB	0.008	−0.157	0.991
DMMP	0.031	−1.138	0.999
	**SAW-3 (TU-2)**		
GA	0.079	−1.527	-
GB	0.011	0.048	0.972
DMMP	0.004	0.485	0.935

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
