# Peer review of "SAW Chemical Array Device Coated with Polymeric Sensing Materials for the Detection of Nerve Agents"

_sensors, 2020, doi:10.3390/s20247028_

Round 1
Reviewer 1 Report
The paper is well written, and well presented.
The methodology is presented in a clear, understandable way.
The experiments were thoroughly conducted. They yielded clean results. Fig. 10 provides interesting information about the evolution of the sensor's performance over time.
*L73: citations [26,27] inadequate (self-citations?). The SAW technology is 55 years old. Please cite one more appropriate, seminal paper.
The utilization of SAW chemical sensors for the detection of vapors and gases is, however, a pretty old topic. All these sensors suffer from the same limitations: cross-sensitivity (i.e. the sensors are actually sensitive to almost everything, which makes it difficult to use them for practical applications), recovery/reproducibility issues (i.e. the sensor doesn't come back to zero, when exposed to air only) and limited sensitivity (hardly better than 1ppm). It looks like the sensors presented in this paper are of the exact same kind of the numerous sensors presented in the numerous papers published on this topic, in the last 20-25 years... and they suffer from the exact same limitations.
It might be nice to add something about these classical limitations, and solutions envisaged to overcome these issues (future work?). Also, how could SAW sensors compete against optical sensors? Especially, for the accurate and reproducible detection of highly dangerous gases/vapors?
One could also ask for more classical papers to be cited here. To cite recent papers gives a false impression of novelty, although similar studies have been conducted and published in the past (please check IEEE IUS Papers, especially).
Author Response
Response to reviewers’ comments
First of all, the author thanks for the reviewers’ valuable comments. Based on the comments, the revision is prepared and the corrections (highlighted in the revision) are listed below point by point:
Reviewer 1
Comments and Suggestions for Authors
The paper is well written, and well presented.
The methodology is presented in a clear, understandable way.
The experiments were thoroughly conducted. They yielded clean results. Fig. 10 provides interesting information about the evolution of the sensor's performance over time.
*L73: citations [26,27] inadequate (self-citations?). The SAW technology is 55 years old. Please cite one more appropriate, seminal paper.
==>Answer: The authors checked and updated the all references and the citation in the revision.
The utilization of SAW chemical sensors for the detection of vapors and gases is, however, a pretty old topic. All these sensors suffer from the same limitations: cross-sensitivity (i.e. the sensors are actually sensitive to almost everything, which makes it difficult to use them for practical applications), recovery/reproducibility issues (i.e. the sensor doesn't come back to zero, when exposed to air only) and limited sensitivity (hardly better than 1ppm). It looks like the sensors presented in this paper are of the exact same kind of the numerous sensors presented in the numerous papers published on this topic, in the last 20-25 years... and they suffer from the exact same limitations.
It might be nice to add something about these classical limitations, and solutions envisaged to overcome these issues (future work?). Also, how could SAW sensors compete against optical sensors? Especially, for the accurate and reproducible detection of highly dangerous gases/vapors?
==> Answer: Currently the other work based on optical sensing system in my lab is under testing. Therefore, the authors will publish the extended or some novel results in near future. The authors appreciate you for valuable comment.
One could also ask for more classical papers to be cited here. To cite recent papers gives a false impression of novelty, although similar studies have been conducted and published in the past (please check IEEE IUS Papers, especially).
==> Answer: Based on comments, we updated the references and added 2 more referenced in the revision.
Reviewer 2 Report
- the abstract should be written in a more coincise way: some abbrevations like GA, GB are not depicted yet
- p.2/48 there can be also the SAW sensor based on conductivity changes as a major effect
- the eq.1 is well known, so it would be better to include into the text?
- could you explain the role of heat chamber on fig4?
- and bubbler in fig5 - is it humidity generator?
- In fig.6 the polymer material should be indicated direct on the figure and not only in a caption also the unit of concentration should be close to their number - pls. give also the flows values - is it all for the same value of the gas flows?
- figs 7 and 8 similar as for 6
- fig.9 also could be better depicted - the names of polymers on the figure - the linear parameters would be better in the form of table?
- in fig.10 please add that it is for DMMP - direct on the figure
- the conclusions could be written in a more coincise way
Author Response
Response to reviewers’ comments
First of all, the author thanks for the reviewers’ valuable comments. Based on the comments, the revision is prepared and the corrections (highlighted in the revision) are listed below point by point:
Comments and Suggestions for Authors
1. the abstract should be written in a more concise way: some abbreviations like GA, GB are not depicted yet
- Answer: As indicated, the abbreviations of GA, GB in the abstract was added as sarin and tabun in the revision.
2. p.2/48 there can be also the SAW sensor based on conductivity changes as a major effect
- Answer: The authors appreciate the reviewer’s comment. A conductivity change, one of the gas sensing mechanisms of SAW sensors, was mentioned and the related seminal paper was cited.
3. the eq.1 is well known, so it would be better to include into the text?
- Answer: As suggested, the associated formula was included into the text. Thank you for suggestion.
4. could you explain the role of heat chamber on fig4?
- Answer: The heat chamber is a heater which can control the temperature of bubbler to vaporize the GA and GB liquid. The chamber condition was maintained at a constant temperature depending on the type of agent (140 °C for GB, 175 °C for GA, which were recommended from TNO, Netherland), these agents in the syringe was injected to the heat chamber with a constant pressure maintained by the syringe driver. The injected agent evaporates in the heat chamber and moves nitrogen outward into the carrier gas.
5. and bubbler in fig5 - is it humidity generator?
- Answer: A bubbler is small vessel which can vaporize DMMP liquid using the carrier gas. Pure nitrogen gas as a carrier enters into the bottom of a DMMP-containing vessel through the inlet pipe, and then takes away the DMMP vapor upward. Because of this, the carrier gas can completely saturate with DMMP vapor.
6. In fig.6 the polymer material should be indicated direct on the figure and not only in a caption also the unit of concentration should be close to their number - pls. give also the flows values - is it all for the same value of the gas flows?
- Answer: The Fig.6 was modified by following the reviewer’s comment. We added the name of polymer materials, and the unit of concentrations was located close to their number. In case of GA and GB, flow rate is 1 L/min, in case of DMMP, the flow rate is 3 L/min.
7. figs 7 and 8 similar as for 6
- Answer: The Fig.7 and 8 were also updated similar to Fig.6.
8. fig.9 also could be better depicted - the names of polymers on the figure - the linear parameters would be better in the form of table?
- Answer: The name of polymer was added in Fig.9 and the parameters appear in Fig.9. Also the detailed number is listed in Table 1.
9. in fig.10 please add that it is for DMMP - direct on the figure
- Answer: The words indicating exposure to DMMP was added in Fig.10
10. the conclusions could be written in a more coincise way
- Answer: The conclusion was revised in the revision. Some sentences were moved the result part. The authors appreciate you for your valuable comment.